# Strongly Incremental Constituency Parsing with Graph Neural Networks

**Kaiyu Yang**
Princeton University
kaiyuy@cs.princeton.edu

**Jia Deng**
Princeton University
jiadeng@cs.princeton.edu

## Abstract

Parsing sentences into syntax trees can benefit downstream applications in NLP. Transition-based parsers build trees by executing actions in a state transition system. They are computationally efficient, and can leverage machine learning to predict actions based on partial trees. However, existing transition-based parsers are predominantly based on the shift-reduce transition system, which does not align with how humans are known to parse sentences. Psycholinguistic research suggests that human parsing is *strongly incremental*—humans grow a single parse tree by adding exactly one token at each step. In this paper, we propose a novel transition system called *attach-juxtapose*. It is strongly incremental; it represents a partial sentence using a single tree; each action adds exactly one token into the partial tree. Based on our transition system, we develop a strongly incremental parser. At each step, it encodes the partial tree using a graph neural network and predicts an action. We evaluate our parser on Penn Treebank (PTB) and Chinese Treebank (CTB). On PTB, it outperforms existing parsers trained with only constituency trees; and it performs on par with state-of-the-art parsers that use dependency trees as additional training data. On CTB, our parser establishes a new state of the art. Code is available at `https://github.com/princeton-vl/attach-juxtapose-parser`.

## 1 Introduction

Constituency parsing is a core task in natural language processing. It recovers the syntactic structures of sentences as trees (Fig. 1). State-of-the-art parsers are based on deep neural networks and typically consist of an encoder and a decoder. The encoder embeds input tokens into vectors, from which the decoder generates a parse tree. A main class of decoders builds trees by executing a sequence of actions in a state transition system [32, 12, 22]. These transition-based parsers achieve linear runtime in sentence length. More importantly, they construct partial trees during decoding, enabling the parser to leverage explicit structural information for predicting the next action.

Most existing transition-based parsers adopt the shift-reduce transition system [32, 12, 22]. They represent the partial sentence as a stack of subtrees. At each step, the parser either pushes a new token onto the stack (`shift`) or combines two existing subtrees in the stack (`reduce`).

Despite their empirical success, shift-reduce parsers appear to differ from how humans are known to perform parsing. Psycholinguistic research [25, 37, 35] has suggested that human parsing is *strongly incremental*: at each step, humans process exactly one token—no more, no less—and integrate it into a single parse tree for the partial sentence. In a shift-reduce system, however, only `shift` actions process new tokens, and the partial sentence is represented as a stack of disconnected subtrees rather than a single connected tree.

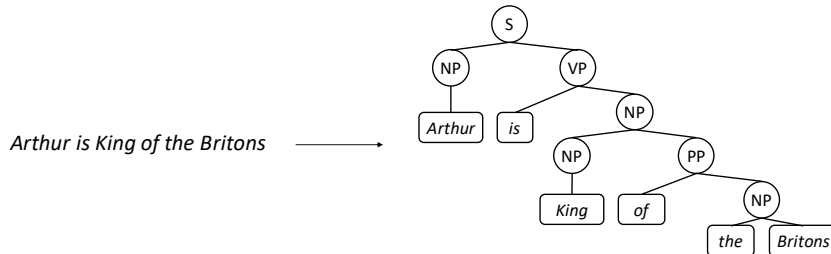

Figure 1: The constituency tree for "Arthur is King of the Britons." Leaves are labeled with tokens, and internal nodes are labeled with syntactic categories, e.g., *S* for *sentence*, *NP* for *noun phrase*, *VP* for *verb phrase*, and *PP* for *prepositional phrase*.

This observation puts forward an intriguing question: can a strongly incremental transition system lead to a better parser? Intuitively, a strongly incremental system is more aligned with human processing; as a result, the sequence of actions may be easier to learn.

**Attach-juxtapose transition system**    We propose a novel transition system named *attach-juxtapose*, which enables strongly incremental constituency parsing. For a sentence of length $n$, we start with an empty tree and execute $n$ actions; each action integrates exactly one token into the current partial tree, deciding on *where* and *how* to integrate the new token. There are two types of actions: `attach`, which attaches the new token as a child to an existing node, and `juxtapose`, which juxtaposes the new token as a sibling to an existing node while also creating a shared parent node (Fig. 2). We can prove that any parse tree without unary chains can be constructed by a unique sequence of actions in this attach-juxtapose system.

Being strongly incremental, our system represents the state as a single tree rather than a stack of multiple subtrees. Not only is the single-tree representation more aligned with humans, but it also allows us to tap into a large inventory of model architectures for learning from graph data, such as TreeLSTMs [39] and graph neural networks (GNNs) [19]. Further, the single-tree representation provides valid syntax trees for partial sentences, which is impossible in bottom-up shift-reduce systems [32]. Taking "Arthur is King of the Britons" as an example, we can produce a valid tree for the prefix "Arthur is King" (Fig. 2 *Bottom*). Whereas in bottom-up shift-reduce systems, you must complete the subtree for "is King of the Britons" before connecting it to "Arthur." Therefore, our representation captures the complete syntactic structure of the partial sentence, and thus provides stronger guidance for action generation.

Our transition system can be understood as a refactorization of In-order Shift-reduce System (ISR) proposed by Liu and Zhang [22]. We prove that a sequence of actions in our system can be translated into a sequence of actions in ISR, but our sequence is shorter (Theorem 4). Specifically, to generate a parse tree with $n$ leaves and $m$ internal nodes (assuming no unary chains), our sequence has length $n$, whereas ISR has length $n + 2m$. On the other hand, each of our actions has a larger number of choices, resulting in a different trade-off between the sequence length and the number of choices per action. We hypothesize that our system achieves a trade-off more amenable to machine learning due to closer alignment with human processing.

**Action generation with graph neural networks**    Based on the attach-juxtapose system, we develop a strongly incremental parser by training a deep neural network to generate actions. Specifically, we adopt the encoder in prior work [21, 49] and propose a novel graph-based decoder. It uses GNNs to learn node features in the partial tree, and uses attention to predict where and how to integrate the new token. To our knowledge, this is the first time GNNs are applied to constituency parsing.

We evaluate our method on two standard benchmarks for constituency parsing: Penn Treebank (PTB) [24] and Chinese Treebank (CTB) [46]. On PTB, our method outperforms existing parsers trained with only constituency trees. And it performs competitively with state-of-the-art parsers that use dependency trees as additional training data. On CTB, we achieve an F1 score of 93.59—a significant improvement of 0.95 upon previous best results. These results demonstrate the effectiveness of our strongly incremental parser.

**Contributions** Our contributions are threefold. First, we propose attach-juxtapose, a novel transition system for constituency parsing. It is strongly incremental and motivated by psycholinguistics. Second, we provide theoretical results characterizing its capability and its connections with an existing shift-reduce system [22]. Third, we develop a parser by generating actions in the attach-juxtapose system. Our parser achieves state-of-the-art performance on two standard benchmarks.

## 2 Related Work

**Constituency parsing** Significant progress in constituency parsing has been made by powerful token representations. Stern et al. [36] fed tokens into an LSTM [17] to obtain contextualized embeddings. Gaddy et al. [14] demonstrated the value of character-level features. Kitaev and Klein [21] replaced LSTMs with self-attention layers [41]. They also show that pre-trained contextualized embeddings such as ELMo [30] significantly improve parsing performance. Further improvements [20, 49] came with more powerful pre-trained embeddings, including BERT [10] and XL-Net [47]. Mrini et al. [27] proposed label attention layers that improved upon self-attention layers. All these works use an existing decoder (chart-based) and focus on designing a new encoder. In contrast, we use an existing encoder by Kitaev and Klein [21] and focus on designing a new decoder.

There are several types of decoders in prior work: chart-based decoders search for a tree maximizing the sum of span scores via dynamic programming (e.g., the CKY algorithm) [36, 14, 21, 49, 27]; transition-based decoders build trees through a sequence of actions [32, 50, 12, 8, 22]; sequence-based decoders generate a linearized sequence of the tree using seq2seq models [42, 5, 38, 15, 23]. Our method is transition-based; but unlike the conventional shift-reduce methods, we propose a novel transition system, which is strongly incremental.

State-of-the-art parsers perform joint constituency parsing and dependency parsing, e.g., through head-driven phrase structure grammar (HPSG) [49, 27]. They use dependency trees as additional training data, which are converted from constituency trees by a set of hand-crafted rules [9]. In contrast, our parser is trained with only constituency trees and achieves competitive performance.

**Transition-based constituency parsers** Most transition-based parsers adopt the shift-reduce transition system: a state consists of a buffer $B$ holding unprocessed tokens and a stack $S$ holding processed subtrees. Initially, $S$ is empty, and $B$ contains the entire input sentence. In a successful final state, $B$ is empty, and $S$ contains a single subtree—the complete parse tree. Below are actions for a standard bottom-up shift-reduce system such as Sagae and Lavie [32], which corresponds to post-order traversal of the complete parse tree.

- `shift`: Remove the first element from $B$ and push it onto $S$.
- `unary_reduce(X)`: Pop a subtree from $S$; add a label X as its parent; and push it back onto $S$.
- `binary_reduce(X)`: Pop two subtrees; add a label X as their shared parent; and push back.

Dyer et al. [12] proposed a top-down (pre-order) variant. Liu and Zhang [22] proposed an in-order shift-reduce system, outperforming bottom-up [32] and top-down [12] baselines. Compared to our transition system, none of these shift-reduce systems is strongly incremental, because they represent the partial sentence as a stack of disconnected subtrees and they do not process exactly one token per action—only the `shift` action consumes a token.

Among the shift-reduce systems, our approach is most related to the in-order system by Liu and Zhang [22] in that each action in our system can be mapped to a combination of actions in their system (see Sec. 3 for details). An analogy is that their actions resemble a set of microinstructions for CPUs, where each instruction is simple but it takes many instructions to complete a task; our actions resemble a set of complex instructions, where each instruction is more complex but it takes fewer instructions to complete the same task.

Transition-based parsers use machine learning to make local decisions—determining the action to take at each step. This poses the question of how to represent a stack of subtrees in shift-reduce systems. Earlier works such as Sagae and Lavie [32] and Zhu et al. [50] use hand-crafted features. More recent works [43, 12, 8, 22] have switched to recurrent neural networks and LSTMs. In particular, Dyer et al. [11] propose an LSTM-based model named Stack LSTM for representing the stack. It is designed for dependency parsing but applies to constituency parsing as well [11, 22]. However, we

do not have to represent stacks thanks to the single-tree state representation. Instead, we use graph neural networks (GNNs) [19] to represent partial trees.

**Incremental parsing**   Prior work has built parsers inspired by the incremental syntax processing of humans. Earlier works focused on psycholinguistic modeling of humans and evaluated on a handful of carefully curated sentences [26, 1]. More recent methods switched to developing efficient parsers with a wide coverage of real-world texts. Roark [31] proposed a top-down incremental parser that expands nodes in the partial tree using a probabilistic context-free grammar (PCFG). In contrast, our system is more flexible by not restricted to any predefined grammar. Also, we predict actions leveraging not only top-down information but also bottom-up information.

Costa et al. [7] proposed a transition system for incremental parsing. Similar to ours, it integrates exactly one token per step into the partial tree. However, at each step, they have to predict an unbounded number of labels, whereas we have to predict no more than two. Therefore, our action space is smaller than theirs and thus easier to navigate by learning-based parsers. In fact, this limitation may have prevented Costa et al. [7] from building a fully functional parser, and they only evaluated on action generation. Collins and Roark [6] developed a parser based on Costa et al. [7] by using grammar rules and heuristics to prune the action space. In contrast, our action space is more flexible without grammar rules but still tractable for machine learning models.

**Graph neural networks for syntactic processing**   GNNs have been used to process syntactic information. These methods obtain syntax trees using external parsers and apply GNNs to the trees for downstream tasks such as pronoun resolution [45], relation extraction [16, 48, 33], and machine translation [3, 4]. In contrast, we apply GNNs to partial trees for the task of parsing. Ji et al. [18] used GNNs for graph-based dependency parsing. However, their method is not transition-based. They apply GNNs to complete graphs formed by all tokens, whereas we apply GNNs to partial trees.

## 3   Attach-juxtapose Transition System

**Overview**   We introduce a novel transition system named *attach-juxtapose* for strongly incremental constituency parsing. Our system is inspired by psycholinguistic research [25, 37, 35]; it maintains a single parse tree and adds one token to it at each step. Our system can produce valid syntax trees for partial sentences and can handle trees with arbitrary branching factors.

For parsing a sentence of length $n$, we start with an empty tree and sequentially execute $n$ actions; each action integrates the next token into the current partial tree. Formally speaking, for the sentence $[w_0, w_1, \ldots, w_{n-1}]$, the state at $i$th step is $s_i = (T_i, w_i)$, where $T_i$ is the partial tree for the prefix $[w_0, \ldots, w_{i-1}]$. The state transition rules are $T_0 = \texttt{empty\_tree}$ and $T_{i+1} = T_i(a_i)$, where $T_i(a_i)$ denotes the result of executing action $a_i$ on tree $T_i$. After $n$ steps, we end up with a complete tree $T_n$. The actions are designed to capture *where* and *how* to integrate a new token into the partial tree.

**Where to integrate the new token**   Since the new token is to the right of existing tokens, it must appear on the *rightmost chain*—the chain of nodes starting from the root and iteratively descending to the rightmost child (A similar observation was also made by Costa et al. [7]). Formally speaking, at the $i$th step, we have a partial tree $T_i$ and a new token $w_i$. Let *rightmost_chain*$(T_i)$ denote the set of *internal nodes* on the rightmost chain of $T_i$. We pick $\texttt{target\_node} \in$ *rightmost_chain*$(T_i)$ as where the new token should be integrated.

**How to integrate the new token**   Fig. 2 *Top* shows the rightmost chain and $\texttt{target\_node}$ (*orange*), we design two types of actions specifying how to integrate the new token (*blue*):

- $\texttt{attach(target\_node, parent\_label)}$: Attach the token as a descendant of $\texttt{target\_node}$. The parameter $\texttt{parent\_label}$ is optional; when provided, we create an internal node labeled $\texttt{parent\_label}$ (*green*) as the parent of the new token. $\texttt{Parent\_label}$ then becomes the rightmost child of $\texttt{target\_node}$ (as in Fig. 2 *Top*). When $\texttt{parent\_label}$ is not provided, the new token itself becomes the rightmost child of $\texttt{target\_node}$.

- $\texttt{juxtapose(target\_node, parent\_label, new\_label)}$: Create an internal node labeled $\texttt{new\_label}$ (*gray*) as the shared parent of $\texttt{target\_node}$ and the new token. It then replaces

`target_node` in the tree. Similar to `attach`, we can optionally create a parent for the new token via the `parent_label` parameter.

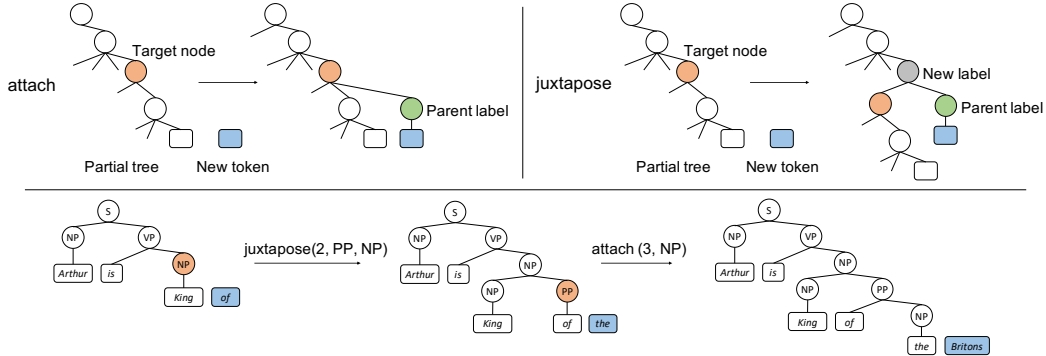

Figure 2: Actions in the attach-juxtapose transition system. *Top-left*: Given a target node (*orange*) on the rightmost chain, the `attach` action attaches the new token (*blue*) as its descendant. *Top-right*: The `juxtapose` action juxtaposes the new token and the target node in different branches of a shared ancestor (*gray*). Both actions can optionally create a parent (*green*) for the new token. *Bottom*: Two example actions when parsing the sentence "Arthur is King of the Britons."

Fig. 2 *Bottom* shows two example actions when parsing "Arthur is King of the Britons." We represent `target_node` using its index on the rightmost chain (starting from 0). The complete action sequence to parse the sentence correctly (Fig. 1) would be: `attach(0, NP)`, `juxtapose(0, VP, S)`, `attach(1, NP)`, `juxtapose(2, PP, NP)`, `attach(3, NP)`, `attach(4, None)`. Note that the first action `attach(0, NP)` is a degenerated case. Since $T_0 = $ `empty_tree`, it is impossible to pick `target_node` $\in$ *rightmost_chain*$(T_0)$. In this case, imagine a dummy root node for $T_0$; then we can execute `attach(0, parent_label)`, making `parent_label` the new root.

**Oracle actions**   Having defined the attach-juxtapose transition system, we are yet to show its capability for constituency parsing: *Given a constituency tree, is it always possible to find a sequence of oracle actions to produce the tree?* This question is important because if the oracle actions did not exist, it would not be possible to parse the sentence correctly. If the oracle actions do exist, a further question is: *For a given tree, is the sequence of oracle actions unique?* Uniqueness is desirable because it guarantees an unambiguous supervision signal at each step when training the parser. We prove that the answers to both questions are positive under mild conditions:

**Theorem 1** (Existence of oracle actions). *Let $T$ be a constituency tree for a sentence of length $n$. If $T$ does not contain unary chains, there exists a sequence of actions $a_0, a_1, \ldots, a_{n-1}$ such that* `empty_tree`$(a_0)(a_1) \ldots (a_{n-1}) = T$.

**Theorem 2** (Uniqueness of oracle actions). *Let $T$ be a constituency tree for a sentence of length $n$, and $T$ does not contain unary chains. If $a_0, a_1, \ldots, a_{n-1}$ is a sequence of oracle actions, it is the only action sequence that satisfies* `empty_tree`$(a_0)(a_1) \ldots (a_{n-1}) = T$.

The condition regarding unary chains is not a restriction in practice, as we can remove unary chains using the preprocessing technique in prior work [21, 49, 27]. The theorems above can be proved by constructing an algorithm to compute the oracle actions. We present the algorithm and detailed proofs in the supplementary materials. Intuitively, given a tree $T$, we recursively find and undo the last action until $T$ becomes `empty_tree`.

**Connections with In-order Shift-reduce System**   Our attach-juxtapose transition system is closely related to In-order Shift-reduce System (ISR) proposed by Liu and Zhang [22]. ISR's state space is strictly larger than ours; we prove it to be equivalent to an augmented version of our state space. Given a sentence $[w_0, w_1, \ldots, w_{n-1}]$, the space of partial trees is $\mathcal{U} = \{t \mid \exists\, 0 \leq m \leq n,$ s.t. $t$ is a constituency tree for $[w_0, w_1, \ldots, w_{m-1}]\}$. By Theorem 1, our state space (for the given sentence) is a subset of $\mathcal{U}$, i.e., $\mathcal{U}_{AJ} = \{t \mid t \in \mathcal{U}, t$ does not contain unary chains$\}$. To bridge $\mathcal{U}_{AJ}$ and $\mathcal{U}_{ISR}$ (the state space of IRS), we define the augmented space of partial trees to be $\mathcal{U}' = \{(t, i) \mid t \in \mathcal{U}, i \in \mathbb{Z}, -1 \leq i < L(t)\}$, where $L(t)$ denotes the number of internal nodes on the rightmost chain of $t$. We assert that $\mathcal{U}'$ is equivalent to $\mathcal{U}_{ISR}$.

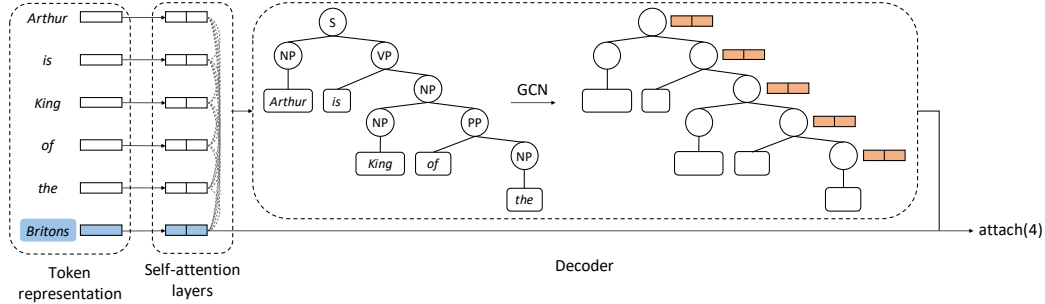

Figure 3: The architecture of our model for action generation. We use the self-attention encoder in prior work [21, 49] and generate actions using a GCN-based [19] decoder.

**Theorem 3** (Connection in state spaces). *There is a bijective mapping $\varphi : \mathcal{U}_{ISR} \to \mathcal{U}'$ between the set of legal states in In-order Shift-reduce System and the augmented space of partial trees.*

We prove the theorem in the supplementary materials. Intuitively, a state in ISR is a stack configuration; it corresponds to an element of $\mathcal{U}'$, which can be understood as a partial tree $t \in \mathcal{U}$ with a special node on the rightmost chain marked by an integer $i$.

Not only is $\varphi$ bijective, but it also preserves actions. In other words, each action in our system can be mapped to a combination of actions in ISR. To see this, we define an injective mapping $\xi : \mathcal{U}_{AJ} \to \mathcal{U}'$ such that $\xi(t) = (t, L(t) - 1)$. Then by Theorem 3, $\varphi^{-1} \circ \xi : \mathcal{U}_{AJ} \to \mathcal{U}_{ISR}$ is an injective mapping from our state space to ISR's state space. And we have the following connection between actions:

**Theorem 4** (Connection in actions). *Let $t_1$ and $t_2$ be two partial trees without unary chains, i.e., $t_1, t_2 \in \mathcal{U}_{AJ}$. If $a$ is an attach-juxtapose action that brings $t_1$ to $t_2$, there must exist a sequence of actions in In-order Shift-reduce System that brings $\varphi^{-1} \circ \xi(t_1)$ to $\varphi^{-1} \circ \xi(t_2)$.*

We present a constructive proof in the supplementary materials, making it possible to translate any action sequence on our system to a longer sequence in ISR. Theorem 4 also implies any reachable parse tree in our system can also be reached in ISR. And by Theorem 1, both our system and ISR can generate any tree without unary chains.

## 4 Action Generation with Graph Neural Networks

Given the attach-juxtapose system, we develop a model for constituency parsing by generating actions based on the partial tree and the new token (Fig. 3). First, it encodes input tokens as vectors using the self-attention encoder in prior work [21, 49]. These vectors are used to initialize node features in the partial tree, which is then fed into a graph convolutional network (GCN) [19]. The GCN produces features for each node, and in particular, the features on the rightmost chain. Finally, an attention-based action decoder generates the action based on the new token and the rightmost chain.

**Token encoder**    We use the same encoder as Kitaev and Klein [21] and Zhou and Zhao [49]. It consists of a pre-trained contextualized embedding such as BERT [10] or XLNet [47], followed by a few additional self-attention layers. Like in prior work, we separate content and position information; the resulting token features are the concatenation of content features and position features. The encoder is not incremental due to how self-attention works; it is applied once to the entire sentence. Then we apply the decoder to generate actions in a strongly incremental manner.

**Graph convolutional neural network**    We use GCN on the partial tree to produce features for nodes on the rightmost chain. Initially, leaf features are provided by the encoder, whereas for an internal node labeled $l$ that spans from position $i$ to $j$ (endpoints included), the initial feature $x = [W_l, x_p] \in \mathbb{R}^D$ is a concatenation of label and position embeddings: $W_l$ is a $\frac{D}{2}$-dimensional learned embedding for label $l$. And $x_p = (P_i + P_j)/2$ is the position embedding averaging two endpoints, where $P$ is the same position embedding matrix in the self-attention encoder.

The initial node features go through several GCN layers with residual connections. We use a variant of the original GCN layer [19] to separate content features and position features (details in the supplementary materials). The GCN produces features for all nodes. However, we are only interested in nodes on the rightmost chain, as they are candidates for `target_node` in actions.

**Action decoder**    Given the structure of our actions (Sec. 3), the action decoder has to (1) choose a `target_node` on the rightmost chain; (2) decide between `attach` and `juxtapose`; and (3) generate the parameters `parent_label` and `new_label`.

We choose `target_node` using attention on the rightmost chain. Let $L$ be the size of the chain, $Y = [Y_c, Y_p] \in \mathbb{R}^{L \times D}$ be the features on the chain produced by the GCN, $z = [z_c, z_p] \in \mathbb{R}^{1 \times D}$ be the feature of the new token given by the encoder. They are both concatenation of content and position features. We generate attention weights for nodes on the rightmost chain as $w = f_c([Y_c, \mathbf{1}^{L \times 1} z_c]) + f_p([Y_p, \mathbf{1}^{L \times 1} z_p])$, where $\mathbf{1}^{L \times 1}$ is a $L \times 1$ matrix of ones. $[\cdot, \cdot]$ concatenates two matrices horizontally, and $f_c$, $f_p$ are two-layer fully-connected networks with ReLU [28] and layer normalization [2]. $w \in \mathbb{R}^{L \times 1}$ and we pick the node with maximum attention as `target_node`.

Since the parameter `new_label` is only for `juxtapose`, we can interpret `new_label = None` as `attach`. So we only need to generate `parent_label`, `new_label` $\in V \cup \{\text{None}\}$, where $V$ is the vocabulary of labels. We generate them using the new token and a weighted average of the rightmost chain: $[u, v] = g([z, \sigma(w)^T Y])$, where $\sigma$ is the sigmoid function, $u, v \in \mathbb{R}^{|V|+1}$ are predicted log probabilities of `parent_label` and `new_label`, and $g$ is a two-layer layer-normalized network.

**Training**    We train the model to predict oracle actions. At any step, let $a = (\texttt{target\_node}, \texttt{parent\_label}, \texttt{new\_label})$ be the oracle action; recall that `new_label = None` implies `attach`, and `new_label` $\neq$ `None` implies `juxtapose`. The loss function is a sum of cross-entropy losses for each component: $\mathcal{L} = \mathcal{CE}(w, \texttt{target\_node}) + \mathcal{CE}(u, \texttt{parent\_label}) + \mathcal{CE}(v, \texttt{new\_label})$. For a batch of training examples, the losses are summed across steps and averaged across different examples.

## 5  Experiments

**Setup**    We evaluate our model for constituency parsing on two standard benchmarks: Chinese Treebank 5.1 (CTB) [46] and the Wall Street Journal part of Penn Treebank (PTB) [24]. PTB consists of 39,832 training examples; 1,700 validation examples; and 2,416 testing examples. Whereas CTB consists of 17,544/352/348 examples for training/validation/testing respectively. Each example is a constituency tree with words and POS tags.

For both datasets, we follow the standard data splits and preprocessing in prior work [22, 34, 21, 49]. In evaluation, we report four metrics—exact match (EM), F1 score, labeled precision (LP), and labeled recall (LR)—all computed by the standard Evalb[1] tool. The testing numbers are produced by models trained on training data alone (not including validation data).

We use the same technique in prior work [21, 49, 27] to remove unary chains by collapsing multiple labels in a unary chain into a single label. It does not affect evaluation, as we revert this process before computing evaluation metrics.

**Training details**    We train the model to predict oracle actions through teacher forcing [44]—the model takes actions according to the oracle rather than the predictions. Model parameters are optimized using RMSProp [40] with a batch size of 32. We decrease the learning rate by a factor of 2 when the best validation F1 score plateaus. The model is implemented in PyTorch [29] and takes $2 \sim 3$ days to train on a single Nvidia GeForce GTX 2080 Ti GPU. The hyperparameters for each model are in the supplementary materials. For fair comparisons with prior work, we use the same pre-trained BERT and XLNet models[2]: `xlnet-large-cased` and `bert-large-uncased` for PTB; `bert-base-chinese` for CTB.

Table 1: Constituency parsing performance on Penn Treebank (PTB). Methods with ⋆ are trained with extra supervision from dependency parsing data. Methods with † are reported in the re-implementation by Fried et al. [13]. Liu and Zhang [22] is transition-based, whereas other baselines are chart-based. We run each experiment 5 times to report the mean and standard error (SEM) of four metrics— exact match (EM), F1 score, labeled precision (LP), and labeled recall (LR). Our method performs competitively with state of the art and achieves the highest EM.

| Model | EM | F1 | LP | LR | #Params |
|---|---|---|---|---|---|
| Liu and Zhang [22] | - | 91.8 | - | - | - |
| Liu and Zhang [22] (BERT) [†] | 57.05 | 95.71 | - | - | - |
| Kitaev and Klein [21] | 47.31 | 93.55 | 93.90 | 93.20 | 26M |
| Kitaev and Klein [21] (ELMo) | 53.06 | 95.13 | 95.40 | 94.85 | 107M |
| Kitaev et al. [20] (BERT) | - | 95.59 | 95.46 | 95.73 | 342M |
| Zhou and Zhao [49] (GloVe) ⋆ | 47.72 | 93.78 | 93.92 | 93.64 | 51M |
| Zhou and Zhao [49] (BERT) ⋆ | 55.84 | 95.84 | 95.98 | 95.70 | 349M |
| Zhou and Zhao [49] (XLNet) ⋆ | 58.73 | 96.33 | 96.46 | 96.21 | 374M |
| Mrini et al. [27] (XLNet) ⋆ | 58.65 | **96.38** | 96.53 | **96.24** | 459M |
| Ours (BERT) | 57.29 ± 0.57 | 95.79 ± 0.05 | 96.04 ± 0.05 | 95.55 ± 0.06 | 377M |
| Ours (XLNet) | **59.17** ± 0.33 | 96.34 ± 0.03 | **96.55** ± 0.02 | 96.13 ± 0.04 | 391M |

**Parsing performance**   Table 1 summarizes our PTB results compared to state-of-the-art parsers, including both chart-based parsers [21, 20, 49, 27] and transition-based parsers [22]. Methods with ⋆ are trained with extra supervision from dependency parsing data. Methods with † are reported in not their original papers but the re-implementation by Fried et al. [13], since the original versions did not use BERT. Liu and Zhang [22] (BERT) [†] performs beam search during testing with a beam size of 10. We do the same for fair comparisons, which improves the performance marginally (0.05 in EM and 0.02 in F1 for our model with XLNet). Some metrics for prior work are missing because they are neither reported in the original papers nor available using the released model and code. We run each experiment 5 times with different random seeds to report the mean and its standard error (SEM). Overall, our method performs competitively with state-of-the-art parsers on PTB. It achieves higher EM using the same pre-trained embedding (BERT or XLNet). Also, our method has a comparable number of parameters with existing methods.

Table 2: Constituency parsing performance on Chinese Treebank (CTB). ⋆ and † bear the same meaning as in Table 1. Our method outperforms state-of-the-art parsers by a large margin.

| Model | EM | F1 | LP | LR |
|---|---|---|---|---|
| Kitaev et al. [20] | - | 91.75 | 91.96 | 91.55 |
| Kitaev et al. [20] (BERT) [†] | 44.42 | 92.14 | - | - |
| Zhou and Zhao [49] ⋆ | - | 92.18 | 92.33 | 92.03 |
| Mrini et al. [27] (BERT) ⋆ | - | 92.64 | 93.45 | 91.85 |
| Liu and Zhang [22] | - | 86.1 | - | - |
| Liu and Zhang [22] (BERT) [†] | 44.94 | 91.81 | - | - |
| Ours (BERT) | **49.72** ± 0.83 | **93.59** ± 0.26 | **93.80** ± 0.26 | **93.40** ± 0.28 |

Table 2 summarizes our results on CTB. Our method outperforms existing parsers by a large margin (0.95 in F1). Compared to PTB, the CTB results have a larger SEM. The reason could be that CTB has a small testing set of only 348 examples, leading to less stable evaluation metrics. However, the SEM is still much smaller than our performance margin with existing parsers.

**Parsing speed**   We measure parsing speed empirically using the wall time for parsing the 2,416 PTB testing sentences. Results are shown in Table 3. It takes 33.9 seconds for our method (with XLNet, without beam search), 37.3 seconds for Zhou and Zhao [49], and 40.8 seconds for Mrini et al. [27]. Our method is slightly faster, but the gap is small. About 50% of the time is spent on the XLNet encoder, which is shared among all three methods and explains their similar run time. These experiments were run on machines with 2 CPUs, 16GB memory, and one GTX 2080 Ti GPU.

Table 3: The wall time for parsing the PTB testing set. We run each experiment 5 times.

|  | Ours (XLNet) | Zhou and Zhao [49] (XLNet) | Mrini et al. [27] (XLNet) |
|---|---|---|---|
| Time (seconds) | **33.9** $\pm$ 0.3 | 37.3 $\pm$ 0.2 | 40.8 $\pm$ 0.9 |

**Effect of the transition system**   Our method differs from Liu and Zhang [22] in not only the transition system but also the overall model architecture. To more closely compare our attach-juxtapose transition system with their In-order Shift-reduce System (ISR), we perform an ablation that only changes the transition system while keeping the other part of the model as close as possible.

We implement a baseline that generates ISR actions on top of our encoder and GCNs. To that end, we rely on Theorem 3 to interpret ISR states (stacks) as augmented partial trees. Specifically, given an ISR state $s \in \mathcal{U}_{ISR}$, we have $\varphi(s) \in \mathcal{U}'$ from Theorem 3. We know $\varphi(s) = (t, i)$, where $t$ is a partial tree and $i$ is an integer ranging from $-1$ to $L(t) - 1$. First, we encode $t$ in the same way as before, using XLNet and GCNs. Then, we take the GCN feature of the $i$th node on the rightmost chain and use it to generate actions in ISR. We add a special node to the rightmost chain to handle $i = -1$.

Results are shown in Table 4. the ISR baseline achieves an average F1 score of 96.23 on PTB, which is lower than our method (96.34). This ablation demonstrates that the attach-juxtapose transition system contributes to the performance.

Table 4: Comparison on PTB between different transition systems. Both models use XLNet for encoding tokens and GCNs for learning graph features.

| Transition system | EM | F1 |
|---|---|---|
| ISR [22] | 58.99 $\pm$ 0.11 | 96.23 $\pm$ 0.04 |
| Attach-juxtapose | **59.17** $\pm$ 0.33 | **96.34** $\pm$ 0.03 |

**Effect of graph neural networks**   A key ingredient of our model is using GNNs to effectively leverage structural information in partial trees. We conduct an ablation study to demonstrate its importance. Specifically, we keep the encoder fixed and replace the graph-based decoder with a simple two-layer network. For each new token, it predicts an action from the token feature alone—no partial tree is built. It predicts `target_node` as an integer in $[0, 249]$. We increase the feature dimensions so that both models have the same number of parameters. Results are in Table 5; the graph-based decoder achieves better performance in all settings, which demonstrates the value explicit structural information.

Table 5: Ablation study comparing our graph-based action decoder with a sequence-based decoder that cannot leverage structural information in partial trees. The graph-based decoder leads to better performance in all settings, which demonstrates the importance of explicit structural information.

| Decode | BERT (PTB) | | XLNet (PTB) | | BERT (CTB) | |
|---|---|---|---|---|---|---|
| | EM | F1 | EM | F1 | EM | F1 |
| Sequence-based | 53.26 $\pm$ 0.45 | 94.89 $\pm$ 0.03 | 55.84 $\pm$ 0.53 | 95.54 $\pm$ 0.07 | 44.14 $\pm$ 0.88 | 90.98 $\pm$ 0.35 |
| Graph-based | **57.29** $\pm$ 0.57 | **95.79** $\pm$ 0.05 | **59.17** $\pm$ 0.33 | **96.34** $\pm$ 0.03 | **49.72** $\pm$ 0.83 | **93.59** $\pm$ 0.26 |

# 6   Conclusion

We proposed the attach-juxtapose transition system for constituency parsing. It is inspired by the strong incrementality of human parsing discovered by psycholinguistics. We presented theoretical results characterizing its capability and its connections with existing shift-reduce systems. Further, we developed a parser based on it and achieved state-of-the-art performance on two standard benchmarks.

## Broader Impact

We evaluated our method on constituency parsing for English and Chinese. They are the two most spoken languages, with more than two billion speakers across the globe. However, there are more than 7,000 languages in the world. And it is important to deliver parsing and other NLP technology to benefit speakers of diverse languages. Fortunately, our method can be applied to many languages with little additional effort. We developed the system using PTB, and when adding CTB, we only had to make a few minor changes in language-specific preprocessing.

However, a potential barrier is the lack of training data for low-resource languages. Our method relies on supervised learning with a large number of annotated parse trees, which are available only for some languages. A potential solution is to do joint multilingual training as in Kitaev et al. [20].

## Acknowledgments and Disclosure of Funding

This work is partially supported by the National Science Foundation under Grant IIS-1903222 and the Office of Naval Research under Grant N00014-20-1-2634.

## Footnotes

[1] `https://nlp.cs.nyu.edu/evalb/`

[2] `https://huggingface.co/transformers/pretrained_models.html`

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
