[Supplementary Material]

# Supplementary Material: Strongly Incremental Constituency Parsing with Graph Neural Networks

**Kaiyu Yang**
Princeton University
kaiyuy@cs.princeton.edu

**Jia Deng**
Princeton University
jiadeng@cs.princeton.edu

## 1 Proofs about Oracle Actions

Given a constituency tree without unary chains, we prove the existence of oracle actions (Theorem 1) by proving the correctness of an algorithm (Algorithm 1) for computing oracle actions. Further, we prove that the oracle action sequence is unique. Before diving into the theorems and proofs, we first define the relevant terms:

**Definition 1** (Constituency tree). *Given a sentence $s = [w_0, w_1, \ldots, w_{n-1}]$ of length $n$, we define a constituency tree for $s$ as a rooted tree with arbitrary branching factors. It has $n$ leaves labeled with tokens $w_0, w_1, \ldots w_{n-1}$ from left to right, whereas its internal nodes are labeled with syntactic categories. The root node must be an internal node. In the degenerated case of $n = 0$, we define a special constant* `empty_tree` *to be the constituency tree for $s = []$.*

**Definition 2** (Unary chain). *Let $T$ be a constituency tree. We say $T$ contains unary chains if there exist two internal nodes $x$ and $y$ such that $y$ is the only child of $x$. Conversely, if such $x$ and $y$ do not exist, we say $T$ does not contain unary chains.*

Then we present Algorithm 1 for computing oracle actions. Given a constituency tree $T$ without unary chains, it recursively finds and undoes the last action until $T$ becomes `empty_tree`. The algorithm has a time complexity of $\mathcal{O}(n \log n)$, where $n$ is the sentence length. It calls *last_action* $n$ times, and each call needs $\mathcal{O}(\log n)$ for locating the last leaf of the tree.

Now we are ready to state and prove Theorem 1 and Theorem 2 in the main paper.

**Theorem 1** (Existence of oracle actions). *Let $T$ be a constituency tree for a sentence of length $n$. If $T$ does not contain unary chains, there exists a sequence of actions $a_0, a_1, \ldots, a_{n-1}$ such that* `empty_tree`$(a_0)(a_1) \ldots (a_{n-1}) = T$. *And this sequence of actions can be computed via Algorithm 1.*

*Proof.* We prove the correctness of Algorithm 1 by induction on the sentence length $n$.

When $n = 0$, we have $T = $ `empty_tree` (Definition 1), and *oracle_action_sequence*$(T)$ returns an empty action sequence $[]$. The conclusion holds straightforwardly.

When $n > 0$, it is sufficient to prove $T'$ is a valid constituency tree without unary chains for a sentence of length $n - 1$. We proceed by enumerating all possible execution traces in *last_action*. The function contains two conditional statements and therfore $2 \times 3 = 6$ execute traces. We use "Case $i$-$j$" to denote the execution trace taking the $i$th branch in the first conditional statement and the $j$th branch in the second conditional statement.

- Case 1-1—*last_leaf* has siblings, and *last_subtree* is the root node.
  We have *last_subtree* = *last_leaf* (the first conditional statement). So *last_leaf* is the root node while being a leaf, which contradicts with the assumption that $T$ is a constituency tree (Definition 1).

**Algorithm 1:** Computing the oracle actions for a constituency tree without unary chains

```
 1  def oracle_action_sequence(T):
 2      if T == empty_tree:
 3          return []
 4      else:
 5          a_{n-1} = last_action(T)
 6          T' = Undo the last action a_{n-1} on T
 7          return oracle_action_sequence(T') + [a]
 8
 9  def last_action(T):
10      last_leaf = The last (rightmost) leaf in T
11      if last_leaf has siblings:
12          parent_label = None
13          last_subtree = last_leaf
14      else:
15          parent_label = The label of last_leaf's parent
16          last_subtree = last_leaf's parent
17
18      if last_subtree is the root node:
19          return attach(0, parent_label)
20      elif last_subtree has exactly one sibling and its sibling is an internal node:
21          new_label = The label of last_subtree's parent
22          target_node = The index of last_subtree's sibling
23          return juxtapose(target_node, parent_label, new_label)
24      else:
25          target_node = The index of last_subtree's parent
26          return attach(target_node, parent_label)
```

Figure A: Case 1-2. The last action is a `juxtapose`.

- Case 1-2—*last_leaf* has siblings; *last_subtree* is not the root node; *last_subtree* has exactly one sibling, and its sibling is an internal node.
  We have *last_subtree* = *last_leaf* (the first conditional statement). The local configuration of $T$ looks like Fig. A *Right*, on its left is $T'$ obtained from $T$ by undoing action juxtapose(*target_node*, *parent_label*, *new_label*). $T'$ is still a valid constituency tree without unary chains.

- Case 1-3—*last_leaf* has siblings; *last_subtree* is not the root node; *last_subtree* has either no sibling, one leaf node as its sibling, or more than one siblings.
  We have *last_subtree* = *last_leaf* (the first conditional statement). So *last_subtree* has either one leaf node or more than one nodes as its siblings. These two cases are shown separately in Fig. B. In both cases, $T'$ is still a valid constituency tree without unary chains.

- Case 2-1—*last_leaf* has no sibling, and *last_subtree* is the root node.
  We have *last_subtree* = *last_leaf*'s parent (the first conditional statement). As shown in Fig. C, $T'$ is empty_tree in this case, which is also a valid constituency tree without unary chains.

- Case 2-2—*last_leaf* has no sibling; *last_subtree* is not the root node; *last_subtree* has exactly one sibling, and its sibling is an internal node.

Figure B: Case 1-3. There are two possible cases depending on the number of siblings of *last_subtree*. In both cases, the last action is an `attach`.

Figure C: Case 2-1. $T'$ = `empty_tree` and *last_subtree* is the root of $T$. The last action is an `attach`.

We have *last_subtree* = *last_leaf*'s parent (the first conditional statement). As Fig. D shows, $T'$ is still a valid constituency tree without unary chains.

- Case 2-3—*last_leaf* has no sibling; *last_subtree* is not the root node; *last_subtree* has either no sibling, one leaf node as its sibling, or more than one siblings.
  We have *last_subtree* = *last_leaf*'s parent (the first conditional statement). So *last_subtree* is an internal node. Since $T$ does not contain unary chains, any non-root internal node must have siblings. As a result, *last_subtree* has either one leaf node or more than one nodes as its sibling. These two cases are shown separately in Fig. E. In both cases, $T'$ is still a valid constituency tree without unary chains.

We have proved $T'$ to be a valid constituency tree for a sentence of length $n-1$ no matter which execution trace *last_action* takes. Applying the induction hypothesis, we know *oracle_action_sequence*($T'$) outputs a sequence of actions $a_0, a_1, \ldots, a_{n-2}$ such that `empty_tree`$(a_0) \ldots (a_{n-2}) = T'$. Since $T'(a_{n-1}) = T$, we have finally derived `empty_tree`$(a_0) \ldots (a_{n-1}) = T$.

$\square$

**Theorem 2** (Uniqueness of oracle actions). *Let $T$ be a constituency tree for a sentence of length $n$, and $T$ does not contain unary chains. If oracle_action_sequence($T$) = $a_0, a_1, \ldots, a_{n-1}$, it is the only action sequence that satisfies* `empty_tree`$(a_0)(a_1) \ldots (a_{n-1}) = T$.

*Proof.* We prove by contradiction. Assume there is different action sequence $a'_0, a'_1, \ldots, a'_{n-1}$ that satisfies `empty_tree`$(a'_0)(a'_1) \ldots (a'_{n-1}) = T$. We first prove $a'_{n-1} = a_{n-1}$, in other words, $a_{n-1}$ is the only possible last action. Similar to Theorem 1, we prove by enumerating all execution traces.

- Case 1-1—*last_leaf* has siblings, and *last_subtree* is the root node.
  Similarly, it contradicts with $T$ being a constituency tree.

- Case 1-2—*last_leaf* has siblings; *last_subtree* is not the root node; *last_subtree* has exactly one sibling, and its sibling is an internal node.

Figure D: Case 2-2. The last action is an `juxtapose`.

Figure E: Case 2-3. There are two possible cases depending on the number of siblings of *last_subtree*. In both cases, the last action is an `attach`.

In Fig. A *Right*, it is apparent that *parent_label* = None. Also, $a'_{n-1}$ must be a `juxtapose` action since otherwise the gray node will introduce an unary chain in $T'$. Therefore, $a'_{n-1} = a_{n-1}$.

- Case 1-3—*last_leaf* has siblings; *last_subtree* is not the root node; *last_subtree* has either no sibling, one leaf node as its sibling, or more than one siblings.
  In Fig. B, *parent_label* = None. No matter how many siblings *last_subtree* has, $a'_{n-1}$ must be an `attach` action. Therefore, $a'_{n-1} = a_{n-1}$.

The remaining three cases are similar, and we omit the details. Now that we have proved $a'_{n-1} = a_{n-1}$, it is straightforward to apply the same reasoning to derive $a'_{n-2} = a_{n-2}$, $a'_{n-3} = a_{n-3}$ and all the way until $a'_0 = a_0$. This contradicts with the assumption that $a'_0, a'_1, \ldots, a'_{n-1}$ is different from $a_0, a_1, \ldots, a_{n-1}$. Therefore, we have an unique sequence of oracle actions $a_0, a_1, \ldots, a_{n-1}$.

$\square$

## 2 Proofs about Connections with In-order Shift-reduce System

We prove Theorem 3 and Theorem 4 in the main paper; they reveal the connections between our system with In-order Shift-reduce System (ISR) proposed by Liu and Zhang [LZ17]. Before any formal derivation, we first illuminate the connections through an example to gain some intuition.

Fig. F shows a `juxtapose` action when parsing "Arthur is King of the Britons." in our system, which can be translated into 4 actions in ISR: `reduce`, PJ-NP, `shift`, PJ-PP. In the figure, $\mathcal{U}_{AJ}$ denotes

Figure F: An `juxtapose` action when parsing "Arthur is King of the Britons." in our attach-juxtapose system. It can be translated into 4 actions in In-order Shift-reduce System.

our state space—the set of partial trees without unary chains. Whereas $\mathcal{U}_{ISR}$ denotes ISR's state space—the set of legal stack configurations. We represent a stack from left (stack bottom) to the right (stack top). "(X)" denotes a projected nonterminal X, while an S-expression such as "(NP Arthur)" denotes a subtree in the stack. $\mathcal{U}'$ denotes the augmented space of partial trees; each element in $\mathcal{U}'$ is a constituency tree that may have a node on the rightmost chain marked as special (*orange*). We observe a one-to-one correspondence between $\mathcal{U}'$ and $\mathcal{U}_{ISR}$, which is denoted by the mapping $\varphi$.

We proceed to generalize this example to arbitrary state transitions in our system, which involves formulating and proving Theorem 3 and Theorem 4 in the main paper. First, we assume a fixed sentence and define the state spaces of our system and ISR:

**Definition 3** (Space of partial trees)**.** *Given the sentence* $[w_0, w_1, \ldots, w_{n-1}]$, *we define the space of partial trees to be:*

$$\mathcal{U} = \{t \mid \exists\, 0 \le m \le n, \text{ s.t. } t \text{ is a constituency tree for } [w_0, w_1, \ldots, w_{m-1}]\}. \quad (1)$$

*From theorem 1, we know that the state space of our system is a subset of* $\mathcal{U}$ *that does not contain unary chains:*

$$\mathcal{U}_{AJ} = \{t \mid t \in \mathcal{U}, t \text{ does not contain unary chains}\}. \quad (2)$$

**Definition 4** (Augmented space of partial trees)**.** *Given the sentence* $[w_0, w_1, \ldots, w_{n-1}]$. *We define the augmented space of partial trees to be:*

$$\mathcal{U}' = \{(t, i) \mid t \in \mathcal{U}, i \in \mathbb{Z}, -1 \le i < L(t)\}, \quad (3)$$

*where* $L(t)$ *denotes the number of internal nodes on the rightmost chain of* $t$. *We can define an injective mapping* $\xi : \mathcal{U}_{AJ} \to \mathcal{U}'$:

$$\xi(t) = (t, L(t) - 1). \quad (4)$$

In ISR, the parser can be trapped in a state that will never lead to a complete tree no matter what actions it takes, e.g., the states where two tokens are at the bottom of the stack. We call such states illegal states and prove a lemma characterizing the set of legal states.

**Lemma 1** (Legal states in ISR)**.** *Let* $s$ *be a stack configuration in In-order Shift-reduce System, it is a legal state if and only if you can end up with a single partial tree in the stack by repeatedly executing* `reduce`.

*Proof.* For the "if" part, we fist keep executing `reduce` until there is only a single partial tree in the stack. Then we can arrive at a complete tree by executing one `PJ-X`, several `shift` to consume all remaining tokens, and one final `reduce`. Therefore, the stack $s$ is legal.

For the "only if" part, $s$ is legal. We prove by contradiction, assuming it is impossible to get a single partial tree by executing multiple `reduce`. Then we must be stuck somewhere. Referring to the definition of the `reduce` action [LZ17], there could be several reasons for being stuck: (1) The stack has more than one element but no projected nonterminal; (2) the only projected nonterminal is at the bottom of the stack; (3) there are two consecutive projected nonterminals. For all three cases, the offending pattern must also exist in the original $s$, and no action sequence can remove them from $s$. Therefore, $s$ must be illegal, which contradicts the assumption. □

Although there are illegal states in ISR, it is possible to avoid them using heuristics in practice. So we do not consider them a problem for ISR. In the following derivations, we safely ignore illegal states and assume ISR's state space to consist of only legal states:

**Definition 5** (Space of legal stack configurations). *Given the sentence $[w_0, w_1, \ldots, w_{n-1}]$, we define $\mathcal{U}_{ISR}$ as the set of legal stack configurations in In-order Shift-reduce System.*

As one of our main theoretical conclusions, ISR's state space is equivalent to the augmented space of partial trees; therefore, it is strictly larger than our state space:

**Theorem 3** (Connection in state spaces). *There is a bijective mapping $\varphi : \mathcal{U}_{ISR} \to \mathcal{U}'$ between the legal states in In-order Shift-reduce System and the augmented space of partial trees.*

*Proof.* For a legal stack $s \in \mathcal{U}_{ISR}$, let $L(s)$ be the number of projected nonterminals in $s$ (introduced by `PJ-X` actions in Liu and Zhang [LZ17]). We are abusing the notation a little bit as we have used $L(t)$ to denote the number of internal nodes on the rightmost chain of a tree $t$. But as we will show, they are actually the same. If $s$ is an empty stack, let $t = $ `empty_tree`. Otherwise, let $t$ be the tree produced by repeatedly executing `reduce` until there is only one partial tree remaining in the stack. This is always possible since $s$ is a legal state (Lemma 1). Then we can define $\varphi(s) = (t, L(s) - 1)$. We prove $\varphi$ is bijective by constructing its inverse mapping $\varphi^{-1} : \mathcal{U}' \to \mathcal{U}_{IRS}$.

Given any $(t, i) \in \mathcal{U}'$, $t$ is a partial tree. We define the depth of a node in $t$ as its distance to the root node. Further, we extend the definition of in-order traversal from binary trees to trees with arbitrary branching factors: the first subtree $\to$ root node $\to$ the second subtree $\to$ the third subtree $\to \ldots$

We define a mapping $\gamma : \mathcal{U}' \to \mathcal{U}_{IRS}$. Let $\gamma(t, i)$ be the stack obtained by starting with an empty stack and traversing the tree $t$ in-order: At any subtree rooted at node $x$, (1) if node $x$ is not on the rightmost chain or $depth(x) > i$, we push the entire subtree $x$ onto the stack and skip traversing the nodes in it. (2) If node $x$ is on the rightmost chain and $depth(x) \leq i$, we push node $x$'s label as a projected nonterminal and keep traversing the nodes in subtree $x$.

We now prove $\gamma$ to be the inverse of $\varphi$, i.e. $\varphi \circ \gamma(t, i) = (t, i)$. It is straightforward that the stack $\gamma(t, i)$ has $i + 1$ projected nonterminals, corresponding to nodes on the rightmost chain with depth $0, 1, \ldots, i$. So, $L(\gamma(t, i)) - 1 = i$, and we only have to prove $t$ to be the tree obtained by repeatedly executing `reduce` on $\gamma(t, i)$.

In the trivial case of $t = $ `empty_tree`, we have $L(t) = 0$, and $i$ must be $-1$. $\gamma(t, i)$ is an empty stack, and executing `reduce` on it will give `empty_tree`, which equals to $t$.

In the non-trivial case of $t \neq $ `empty_tree`, we prove by induction on the number of `reduce` actions ($k$) executed on the stack $\gamma(t, i)$ to get a single tree.

When $k = 0$, $\gamma(t, i)$ contains a single tree, which must be $t$ itself.

When $k > 0$, let $\gamma_1(t, i)$ be the stack after executing one `reduce` on $\gamma(t, i)$. We assert that $\gamma_1(t, i) = \gamma(t, i - 1)$. We can see this by comparing the in-order traversal of $(t, i)$ and $(t, i - 1)$. The only difference is how we process the subtree rooted at the $i$th node on the rightmost chain. When traversing $(t, i)$, we push a projected nonterminal and proceed to nodes in the subtree. When traversing $(t, i - 1)$, we push the entire subtree and skip the nodes in it, which corresponds exactly to executing one `reduce` on $\gamma(t, i)$. Therefore, $\gamma_1(t, i) = \gamma(t, i - 1)$.

We only need $k - 1$ `reduce` actions to get a single tree from the stack $\gamma_1(t, i)$, or equivalently, $\gamma(t, i - 1)$. Applying the induction hypothesis, we will get the tree $t$ by repeatedly applying `reduce` on $\gamma_1(t, i)$. Therefore, we will also get the same $t$ by repeatedly applying `reduce` on $\gamma(t, i)$, i.e. $\varphi \circ \gamma(t, i) = (t, i)$.

Since $(t, i)$ is arbitrary, we have proved $\gamma$ to be the inverse mapping of $\varphi$, and $\varphi$ is thus bijective.

$\square$

**Corollary 1** (Connections in state spaces). *$\varphi^{-1} \circ \xi : \mathcal{U}_{AJ} \to \mathcal{U}_{ISR}$ is an injective mapping from our state space to ISR's state space.*

*Proof.* It is straightforward given that $\xi : \mathcal{U}_{AJ} \to \mathcal{U}'$ is injective (Definition 4) and $\varphi : \mathcal{U}_{ISR} \to \mathcal{U}'$ is bijective (Theorem 3). $\square$

The mapping $\varphi^{-1} \circ \xi$ bridges our state space and ISR's state space. Not only is it injective, but it also preserves actions—each action in our system can be mapped to a combination of actions in ISR. We prove this for `attach` actions (Lemma 2) and `juxtapose` actions (Lemma 3) separately.

**Lemma 2** (Translating `attach` actions to ISR). *Let $t_1$ and $t_2$ be two partial trees without unary chains, i.e., $t_1, t_2 \in \mathcal{U}_{aj}$. If `attach(i, X)` brings $t_1$ to $t_2$, The following action sequence in In-order Shift-reduce System will bring $\varphi^{-1} \circ \xi(t_1)$ to $\varphi^{-1} \circ \xi(t_2)$:*

$$\underbrace{\texttt{reduce}, \ldots, \texttt{reduce}}_{L(t_1)-i-1}, \quad \texttt{shift}, \quad \underbrace{\texttt{PJ-X}}_{\text{if } X \neq \texttt{None}}, \tag{5}$$

*where $L(t_1)$ is the number of internal nodes on the rightmost chain of $t_1$. $X = \texttt{None}$ means the optional argument parent_label is not provided; in this case, we exclude `PJ-X`.*

*Proof.* We know $\varphi^{-1} \circ \xi(t_1) = \varphi^{-1}(t_1, L(t_1) - 1)$ (Definition 4). Recall that in Theorem 3 we have proved that executing one `reduce` on $\varphi^{-1}(t, i)$ gives us $\varphi^{-1}(t, i - 1)$. Therefore, executing `reduce` $L(t_1) - i - 1$ times on $\varphi^{-1}(t_1, L(t_1) - 1)$ gives us $\varphi^{-1}(t_1, i)$. We still have to execute one `shift` and one optional `PJ-X`.

When $X = \texttt{None}$, we only have to execute a `shift`. The resulting stack is $\varphi^{-1}(t_1, i)$ plus a new token at the top. We prove that the new stack equals to $\varphi^{-1}(t_2, L(t_2) - 1)$. Since $t_2$ is a result of executing `attach(i, None)` on $t_1$, we know $L(t_2) - 1 = i$ from the definition of the `attach` action. So, we only have to prove that the new stack equals to $\varphi^{-1}(t_2, i)$. We unfold the definition of $\varphi^{-1}$ (in the proof of Theorem 3) and compare the in-order traversal of $(t_2, i)$ and $(t_1, i)$. When visiting the subtree rooted at *target_node* $i$ in $t_2$, we have the new token as the rightmost child; it corresponds to the new token at the top of the stack. Therefore, $\varphi^{-1}(t_2, i)$ equals to $\varphi^{-1}(t_1, i)$ plus the new token. We have proved the new stack to be $\varphi^{-1}(t_2, L(t_2) - 1)$ and therefore it equals to $\varphi^{-1} \circ \xi(t_2)$ (Definition 4).

When $X \neq \texttt{None}$, we have to execute a `shift` and a `PJ-X`. The resulting stack is $\varphi^{-1}(t_1, i)$ plus a new token and a projected nonterminal $X$ at the top. Similarly, we want to prove the new stack to equal to $\varphi^{-1}(t_2, L(t_2) - 1)$. The reasoning is similar, by comparing the in-order traversal of $(t_1, i)$ and $(t_2, i)$. We thus omit the details.

Therefore, in ISR, we can arrive at the state $\varphi^{-1} \circ \xi(t_2)$ from the state $\varphi^{-1} \circ \xi(t_1)$ by executing $L(t_1) - i - 1$ `reduce` actions, one `shift` action and one optional `PJ-X` action. $\square$

**Lemma 3** (Translating `juxtapose` actions to ISR). *Let $t_1$ and $t_2$ be two partial trees without unary chains, i.e., $t_1, t_2 \in \mathcal{U}_{aj}$. If `juxtapose(i, X, Y)` brings $t_1$ to $t_2$, The following action sequence in In-order Shift-reduce System will bring $\varphi^{-1} \circ \xi(t_1)$ to $\varphi^{-1} \circ \xi(t_2)$:*

$$\underbrace{\texttt{reduce}, \ldots, \texttt{reduce}}_{L(t_1)-i}, \quad \texttt{PJ-Y}, \quad \texttt{shift}, \quad \underbrace{\texttt{PJ-X}}_{\text{if } X \neq \texttt{None}}. \tag{6}$$

*Proof.* Similar to Lemma 2, we first execute $L(t_1) - i$ `reduce` actions on $\varphi^{-1} \circ \xi(t_1) = \varphi^{-1}(t_1, L(t_1) - 1)$ to get $\varphi^{-1}(t_1, i - 1)$. We still have to execute one `PJ-Y`, one `shift` and one optional `PJ-X`.

When $X = \texttt{None}$, we only have to execute a `PJ-Y` and a `shift`. The resulting stack is $\varphi^{-1}(t_1, i - 1)$ plus a projected nonterminal $Y$ and a new token. We prove that the new stack equals to $\varphi^{-1}(t_2, L(t_2) - 1)$. Since $t_2$ is a result of executing `juxtapose(i, None, Y)` on $t_1$, we know $L(t_2) - 1 = i$ from the definition of the `juxtapose` action. So, we only need the new stack to equal to $\varphi^{-1}(t_2, i)$, which can be proved by comparing the in-order traversal of $(t_2, i)$ and $(t_1, i - 1)$: In $t_2$, the subtree rooted at $Y$ has 2 children; the left child corresponds to the $i$th subtree on the rightmost chain of $t_1$, whereas the right child is a single leaf. When we reach $Y$ in the in-order traversal of $t_2$, we first push its left subtree onto the stack. Now the stack equals to $\varphi^{-1}(t_1, i - 1)$. Then, we visit the node $Y$ and its right child, which push the additional projected nonterminal $Y$ and a new token onto the stack. Therefore, $\varphi^{-1}(t_2, i)$ is the new stack.

When $X \neq \texttt{None}$, we have to execute a `PJ-Y`, a `shift`, and a `PJ-X`. The derivation is similar, so we omit the details. $\square$

**Theorem 4** (Connection in actions). *Let $t_1$ and $t_2$ be two partial trees without unary chains, i.e., $t_1, t_2 \in \mathcal{U}_{aj}$. If $a$ is an attach-juxtapose action that brings $t_1$ to $t_2$, there must exist a sequence of actions in In-order Shift-reduce System that brings $\varphi^{-1} \circ \xi(t_1)$ to $\varphi^{-1} \circ \xi(t_2)$.*

*Proof.* It follows straightforwardly from Lemma 2 and Lemma 3. □

**Corollary 2** (Connection in action sequences). *Let $t_1$ and $t_2$ be two partial trees without unary chains, i.e., $t_1, t_2 \in \mathcal{U}_{aj}$. If there exists a sequence of attach-juxtapose actions that brings $t_1$ to $t_2$, there must exist a sequence of actions in In-order Shift-reduce System that brings $\varphi^{-1} \circ \xi(t_1)$ to $\varphi^{-1} \circ \xi(t_2)$.*

*Proof.* It is straightforward to prove using Theorem 4 and induction on the number of actions for bringing $t_1$ to $t_2$. Details are omitted. □

## 3 Separating Content and Position Features in GCN Layers

We use the encoder in Kitaev and Klein [KK18]. They showed that separating content and position features improves parsing performance. Specifically, the input token features to the encoder are the concatenation of content features (e.g., from BERT [DCLT19] or XLNet [YDY+19]) and position features from a learnable position embedding matrix $P$. Kitaev and Klein [KK18] proposed a variant of self-attention that processes two types of features independently. As a result, the output token features are also the concatenation of content and position.

We extend this idea to GCN layers. Vanilla GCN layers perform a linear transformation on the input node feature (i.e. $y = \Theta x + b$) before normalizing and aggregating the neighbors[1]. We assume the node features to be concatenation of content and position: $x = [x_c, x_p]$, and perform linear transformations for them separately: $y = [y_c, y_p] = [\Theta_c x + b_c, \Theta_p x + b_p]$. As a result, the node features at each GCN layer are also concatenation of content and position. As stated in the main paper, we merge the two types of features when generating attention weights $w$ for nodes on the rightmost chain.

To study the effect of the separation, we present an ablation experiment. We compare our model with vanilla GCN layers and with GCN layers that separate content and position. We also scale the feature dimensions so that both models have approximately the same number of parameters. Results are summarized in Table A. We run each experiment 5 times and report the mean and its standard error (SEM). For PTB, we use XLNet as the pre-trained embeddings. Results show that separating content and position improves performance in all settings, which is consistent with prior work [KK18].

Table A: Ablation study separating content and positions.

| GCN layer | PTB | | CTB | |
|---|---|---|---|---|
| | EM | F1 | EM | F1 |
| Vanilla GCN [KW17] | $58.90 \pm 0.23$ | $96.25 \pm 0.06$ | $49.54 \pm 0.65$ | $93.49 \pm 0.27$ |
| Separating content and position | $\mathbf{59.17} \pm 0.33$ | $\mathbf{96.34} \pm 0.03$ | $\mathbf{49.72} \pm 0.83$ | $\mathbf{93.59} \pm 0.26$ |

## 4 Error Categorization using Berkeley Parser Analyzer

We use Berkeley Parser Analyzer [KHCK12] to categorize the errors of Mrini et al. [MDB+20] and our model (with XLNet) on PTB. Results are shown in Table B. Two methods have the same relative ordering of error categories. The 3 most frequent categories are "PP Attachment", "Single Word Phrase", and "Unary". Compared to Mrini et al., our method has more "PP Attachment" (342 vs. 320) and "UNSET move" (33 vs. 23), but fewer "Clause Attachment" (110 vs. 122) and "XoverX Unary" (48 vs. 56).

Table B: Categorization of parsing errors made by Mrini et al. [MDB+20] and our model (with XLNet). For each category, we show its occurrence and the number of brackets attributed to it.

| Error category | Mrini et al. [MDB+20] | | Ours (XLNet) | |
|---|---|---|---|---|
| | #Errors | #Brackets | #Errors | #Brackets |
| PP Attachment | **320** | 746 | 342 | 804 |
| Single Word Phrase | 267 | 325 | **259** | 324 |
| Unary | 237 | 237 | **235** | 235 |
| NP Internal Structure | **200** | 230 | 202 | 236 |
| Different label | **197** | 394 | **197** | 394 |
| Modifier Attachment | 137 | 251 | **131** | 283 |
| Clause Attachment | 122 | 398 | **110** | 334 |
| UNSET add | **85** | 85 | 88 | 88 |
| UNSET remove | 78 | 78 | **71** | 71 |
| Co-ordination | 68 | 138 | **65** | 120 |
| XoverX Unary | 56 | 56 | **48** | 48 |
| NP Attachment | 44 | 160 | **39** | 135 |
| UNSET move | **23** | 62 | 33 | 113 |
| VP Attachment | **12** | 37 | 17 | 56 |

## Footnotes

[1] `https://pytorch-geometric.readthedocs.io/en/latest/notes/create_gnn.html`