[Reviews · NeurIPS 2020]

Review 1

Summary and Contributions: This paper proposes a novel transition-based constituency parsing algorithm, which they call “Attach-juxtapose transition system”. The proposed method consists of two actions, attach and juxtapose. Besides, this paper also proposes to use Graph convolutional neural networks (GNNs) as the scoring function for selecting actions for each step. The experimental results on the English and Chinese constituency parsing show that the proposed method outperformed the current SOTA reported in the previous studies. Moreover, this paper also demonstrates several theoretical analyses to show the characteristics and behaviors of the proposed method.

Strengths: 1, This paper proposes a novel transition-based parsing framework called the Attach-juxtapose transition system. As far as I know, this parsing system is a novel parsing system that is first introduced in this paper. 2, The proposed method achieves the best scores of parsing experiments on English and Chinese PTBs. Achieving SOTA results is not essential for the acceptance of this paper to the conference but can have a positive impact on the community. 3, This paper provides the theoretical proof to obtain the oracle action sequences from the existent parse trees. 4, This paper shows the connections between the proposed method and in-order shift-reduce systems.

Weaknesses: The following are my concerns (questions) and confirmations of the proposed method. 1, This paper does not provide any theoretical analysis of the algorithm in terms of efficiency or actual runtime. I understand that the number of actions required to parse a sentence for the proposed method is $n$, where the number of tokens in the sentence is $n$. However, the computational cost for one action seems relatively very expensive comparing with the existing transition-based algorithm, such as standard shift-reduce parser. Therefore, I suspect that the actual runtime of parsing a single sentence takes much larger than the conventional methods. Regardless of my suspicion is correct or not, the experimental results in the current version does not answer this point. ====> after rebuttal period: Thank you for reporting the actual runtime. However, I am not totally satisfied with the answer. This is because there is no discussion of why the proposed algorithm can run fastest among the comparative methods. Please also add such a discussion or analysis in the camera-ready version if this paper is accepted to the conference. 2, Table 3 empirically shows the apparent effectiveness of the graph-based scoring method, and it also shows the significant degradation if we select a standard sequence-based scoring method. This fact may suggest that the dominant part for improvements by the proposed method comes from the GNN scoring method, not the proposed attach-juxtapose transition system. If this interpretation is correct, then one of the central claims of this paper, namely, attach-juxtapose parsing algorithm is beneficial is not supported by the experiments in this paper. Then, this can be a clear risk to mislead the readers to the effectiveness of the proposed transition-based system. ====> after rebuttal period: The authors provided the results of additional ablation studies in their rebuttal. I found the relation that GNN: 95.54 \pm 0.07 => 96.34 \pm 0.03 attach-juxtapose: 96.24 \pm ?.?? => 96.34 \pm 0.03 These results clearly indicate that the dominant part of the performance gain comes from GNN. I believe that the central claim of this paper is the proposed parsing algorithm, NOT the way of applying GNN. I seriously concern this inconsistency. I never try to say that the proposed parsing algorithm itself is meaningless, but the potential risk by the inaccurate statements that may easily imagine wrong conclusions to many readers is problematic. 3, This paper only shows the improvements of overall performance but provides no detailed analyses of why the proposed method can essentially improve the performance. ====> after rebuttal period: I believe that the Berkeley parser Analysis provided in the rebuttal is crucially informative to readers. I understand that the authors only show its partial result because of the limited space of rebuttal. I would like authors to show all the detailed results in the revised version (maybe in the Appendix section) and discuss the characteristics and behaviors by linking to the theoretical properties that the proposed algorithm has. 4, The broader impact written in this paper is not thoroughly convincing to me. Since the parsing is not the primary application of the ML/NLP research these days, the proposal of the novel parsing system may have a limited impact on the community. ====> after rebuttal period: The authors’ rebuttal does not entirely convince me, unfortunately. If we were in the before neural era, I mostly agree with the potential impact of the proposed method (or incremental parsing). However, nowadays, the importance of incremental parsing is reducing significantly because of the great success of the sequential process of neural language models and end-to-end applications. Of course, shortly, the importance of incremental parsing may increase, hopefully, but no one knows currently, I believe. Anyhow, I decline this part from the main weakness of the paper since it is hard to say which is correct.

Correctness: The main claim of this paper seems valid and correct except one. The discussion of “strongly incremental” in this paper is not enough to support the correctness of this assumption. Therefore, the requirement of “strongly incremental” parser is still unclear to me. ====> after rebuttal period: Unfortunately, the authors did not answer this part. This part is also crucial in the paper, and thus the authors need to appropriately resolve this unclearness since this is the premise of the proposed method. Therefore, this is one of the clear drawbacks of the paper.

Clarity: The paper is basically well-written, and most part is understandable.

Relation to Prior Work: The relate work and explaining the related work seem sufficient.

Reproducibility: Yes

Additional Feedback: I am willing to increase my score if I misunderstand something, or the authors provide reasonable answers.


Review 2

Summary and Contributions: This paper is about building a "strongly incremental" constituent parser, in which the parse tree is constructed while reading in one new token at a time. The core approach is a transition-based parser, where action generation is implemented as a graph convolutional neural network. The paper reports state-of-the-art results on common benchmark datasets.

Strengths: A strength of the work is its state-of-the-art empirical results on two well-studied benchmark datasets. I wonder, however, whether comparing against parse trees from the PTB and the CTB are the right experiments. Since the motivation of the proposed approach seems to stem from psycholinguistics, perhaps a better domain would have been spoken sentences (e.g., Switchboard). The ablation study suggested GNN was helpful for representing structural information. However, the experiment does not reveal deeper insights about GNN and representing structural information.

Weaknesses: The contribution of the work feels somewhat limited. While the experimental results are positive, they did not seem especially surprising or impactful. The scope of the work presented seems rather narrow.

Correctness: The empirical methodology appears to be quite reasonable.

Clarity: The paper is well written.

Relation to Prior Work: The paper does a good job in terms of comparing against relevant recent literature.

Reproducibility: Yes

Additional Feedback: In its present form, the scope of the paper seems too narrow. It is also somewhat unclear whom the intended audience ought to be. If the work aims to say something about psycholinguistics, the experiment should reflect that. If the work's goal is to support NLP applications, further justifications and motivations should be provided as to how a strongly incremental constituency parser might be useful in a current NLP pipeline. If the work aims to shed lights on our understanding of GNN, the paper would need to be refocused accordingly. ** I have considered the authors' response as well as the opinions of the other reviewers; as a result, I have raised the overall score slightly, but I still maintain that the paper could be better motivated.


Review 3

Summary and Contributions: The paper proposes an incremental neural network-based constituency parser that is based on a variant of the previously proposed in-order transition system. The high-level differences are - Using attention to decide where in the right-most chain to attach the next word, rather than a sequence of reduce actions as is typical in transition-based parsers. The result is a strongly incremental parser that has psycho-linguistic motivation. - Encoding the partial tree with a graph convolutional neural network (in addition to self-attention for contextual token embeddings). This is shown to improve performance substantially over sequential-only embedding (the previous in-order approach used a stack LSTM). Results show that for the English PTB the parser performs marginally better than the previous in-order approach when both are based on BERT embeddings, and on par with state-of-the-art performance with XLNet embeddings. On the Chinese CTB it outperforms both the in-order parser and state-of-the-art performance. -- Thanks for the clarifications, in particular the additional experiments. I really want to like this paper, but unfortunately both the performance gains and conceptual advantages of the proposed approach is really small.

Strengths: - Proposes a novel constituency parser with state-of-the-art performance in two languages. The parser has psycho-linguistic motivation. - The proposed model gives a more expressive parameterization of fully incremental parsing than previous approaches. - The paper is very well-written and well-structured.

Weaknesses: - The approach is not that novel compared to previous incremental parsing approaches. The main controbution is a more expressive parameterization of parsing action prediction. - The results are not significantly higher than that of the closest previous work on English. On Chinese it is, but one has to wonder whether that might be due to implementation or hyperparameter differences rather than the particular model. - There are some missing ablations for quantifying where the performance gains come from, and for justifying some design choices (see details below).

Correctness: The claims and the empirical methodology are correct. The comparison with the previous in-order transition system could be improved by having an ablation that uses that instead of attention-based attachment, without changing anything else in the model.

Clarity: The paper is very well written.

Relation to Prior Work: The relation to prior work is clear. One aspect that can be strengthened is the relation with Liu and Zhang: They use a StackLSTM to encode the context, while this work uses a GCN. An example on the relation between the transition systems in the main paper would also improve clarity.

Reproducibility: Yes

Additional Feedback: - In the sequence-based ablation, how is the attention over nodes in the right-most chain computed? The paper (line 310) states that only the token feature is used --- does that mean that no attention is used to decide the attachment? A potentially stronger baseline would be to use the token encoding corresponding to the start of the tokens spanned by the node for computing the attention (along with the node label and positional encoding as used in the GCN), similar to span-based representations used in neural chart parsers. - Would it not make more sense to have separate parameterizations of the attention mechanism over target nodes for the attach and juxtapose actions, and make a joint decision, rather than predicting the target node before choosing the action? With the same target node, the two actions attach the new token on a different level, so in the case of ambiguity this might lead to avoidable suboptimal decisions. - This approach could also be applied to dependency parsing, as in previous work such as Liu and Zhang. - It would help to have a high-level explanation of when the oracle uses attach and when juxtapose. - One potential advantage of this approach is parsing speed, so it would be interesting to compare it to previous approaches here. - It would be interesting to apply the model as a purely incremental parser in which only the left context is considered when making the attachment decision of the next token, rather than the full input sentence - i.e., using a unidirectional rather than a bidirectional transformer.


Review 4

Summary and Contributions: The authors present a novel transition-based constituency parsing system that achieves competitive results with state-of-the-art methods in English and Chinese. The unique properties are: (a) action sequence O(n), which is short considering other transition-based parsers are often O(n+m) or worse, where n is number of tokens and m is number of internal nodes, (b) the intermediate state a time step t is a partial parse of the first t tokens, and (c) the intermediate state is represented using a GCN of the partial parse.

Strengths: The proposed method works well and is straightforward with convenient properties over similar methods and is motivated well from psycholinguistics. Ablation indicates that encoding the entire partial parse at each step is more beneficial than encoding nodes separately. --- After rebuttal: Although the connection to psycholinguistics is appealing, I would encourage the authors to downplay this in the story of the work as this is primarily a point made in the ISR paper. I assume the connection to psycholinguistics is there only because attach-juxtapose maps directly to ISR.

Weaknesses: The extent of ablations done is limited, and leaves the reader with questions: Does choice of GCN impact performance? Is GCN necessary, or are other tree encoders equally useful? --- After rebuttal: The authors ran some new experiments for their rebuttal. I find these to be just as strong as any experiments that would have been done to measure strength of tree encoder (i.e. GCNs), and believe that exploring different tree encoders could be left as future work.

Correctness: The experiments seem to be carried out in a fair way to compare with existing work.

Clarity: Yes the paper is well written and easy to follow.

Relation to Prior Work: The treatment of existing parsers is clear and covers the main competing work. Perhaps some more discussion of parsers that model intermediate states and comparison of action spaces could be done, but is not as critical.

Reproducibility: Yes

Additional Feedback: Very excellent paper. The results are strong and the modeling changes are interesting, useful. That being said, readers could benefit from more ablation and analysis. # questions about error analysis Have you considered further analysis of model output using a tool such as berkeley parser analyser (Kummerfeld et al.)? # questions about training and inference Did you consider a structured or global loss? Is it surprising that beam search helps only marginally overall? Is this because the original prediction is strong, or because beam search helps and hurts in roughly equal amounts? # questions about action space How important is the change in action space compared to modeling decisions? Have you considered training a simple seq2seq model to predict your action space and compare it with an alternative one (i.e. bottom shift reduce actions). Along these lines, how important is it to use BERT? Methods without pre-training on external data can achieve low to mid 90s in labeled parsing F1. # typos Line 280 We use the same technique in -> add “as” # More related work The authors may be interested in the SPINN model from Bowman et al. It is a shift-reduce parser that also encodes the sequence of actions with a separate state from the subtrees on the stack. The incremental parser by Cai and Lam has impressive results for AMR parsing (was not evaluated for constituency parsing). Also, the stack-transformer (Anonymous) models the parser state using dedicated attention heads. Kummerfeld et al. https://www.aclweb.org/anthology/D12-1096/ Bowman et al. https://arxiv.org/abs/1603.06021 Cai and Lam https://arxiv.org/abs/2004.05572 Anonymous https://openreview.net/forum?id=b36spsuUAde --- After rebuttal: I found the authors address most of the reviewer comments in their rebuttal, including discussion of parser speed, comparison of ISR to attach juxtapose, and ran multiple baselines/experiments. Based own my own comments and from the other reviewers, there is still some area for improvement such as a) more detailed runtime complexity analysis (perhaps can explain why the model is faster/slower than any competitors as a minimum addition to the text); b) explain more clearly the results and especially what the main contribution is, and consider making story of this work more about representing partial trees in supervised parsing; c) parts of the paper (especially section 3 and 4) read too much like a technical report, and it may be beneficial to include more justification for each of the design choices. In general I still believe this is a strong work even with these critiques, and that the research community would benefit greatly from this submission.

[Author Response · NeurIPS 2020]

We thank the reviewers for their thoughtful comments. We are delighted that reviewers unanimously find our work novel, well-written, and state-of-the-art on well-studied benchmarks. Most of their questions ask for additional analysis and ablation experiments. Below we try to provide as many of them as we could accomplish within the tight time frame.

**R1, R3, R4: Where does the performance gain come from—the transition system or the GNN?** We perform an ablation study that only changes the transition system while keeping GNNs untouched. We implement In-order Shift-reduce System (ISR) [22]. In order to apply GNNs to ISR, we rely on Theorem 3 to interpret ISR states (stacks) as augmented partial trees. ISR + GNNs (with XLNet encoder) achieves an F1 score of 96.24 on PTB, which is lower than our method ($96.34 \pm 0.03$). This ablation demonstrates that the attach-juxtapose transition system contributes to the performance.

**R4: Does the choice of GNNs matter? Besides GNNs, what about other tree encoders?** We experimented with TreeLSTMs [39] and multiple GNN architectures such as GATs [Veličković et al. ICLR 2018]. Some of them were trained faster than GCNs, but they converged to a similar final performance. We prefer GNNs to TreeLSTMs because they work for graphs with loops. Although our graph is a tree without loops, variants of our method could violate the tree constraint by introducing additional edges,e.g., between consecutive tokens. We chose GCNs among other GNNs because it is straightforward to separate content and position information (details in the supplementary material).

**R1, R4: Detailed analysis of the performance, perhaps using Berkeley Parser Analyser.** We use Berkeley Parser Analyzer [Kummerfeld et al. EMNLP 2012] to categorize the errors of Mrini et al. [27] and our model on PTB. Two methods have the same relative ordering of error categories. The 3 most frequent categories are "PP Attachment", "Single Word Phrase", and "Unary". Compared to Mrini et al., our method has more "PP Attachment" (342 vs. 320) and "UNSET move" (33 vs. 23), but fewer "Clause Attachment" (110 vs. 122) and "XoverX Unary" (48 vs. 56). We will present the detailed results in the revised paper.

**R1, R3: Compare with existing parsers in terms of efficiency.** Our method is slightly faster than existing parsers, measured by the wall time for parsing the 2,416 PTB testing examples. It takes $33.9 \pm 0.3$ seconds for our method (with XLNet, without beam search), $37.3 \pm 0.2$ seconds for Zhou and Zhao [49], and $40.8 \pm 0.9$ seconds for Mrini et al. [27]. About 50% of the time is spent on the XLNet encoder, which is the same computation for all three methods. We run these experiments on machines with 2 CPU cores, 16GB memory, and one Nvidia GeForce GTX 2080 Ti GPU.

**R3: Additional baselines.** We experiment with the two baselines suggested by R3. As the first baseline, we compute attention for the rightmost chain using token embeddings at each node's starting position. It achieves an F1 score of 96.18 on PTB, which is between the sequence-based ablation ($95.54 \pm 0.07$) and our method ($96.34 \pm 0.03$). As the second baseline, we compute attentions for `attach` and `juxtapose` actions separately. It achieves an F1 score of 96.35 on PTB. It is not clear whether this is better or worse than our original method (96.35 vs. 96.34 $\pm 0.03$). However, we will investigate more closely and include the results in the revised paper.

**R3: Limited novelty compared to previous incremental parsers. The contribution is a more expressive parameterization of parsing action prediction.** We respectfully disagree that our contribution is "a more expressive parameterization." Compared to the closest incremental parser (Collins and Roark [6]), our main novelty is in the action space itself, rather than how it is parameterized. Our actions can produce any valid tree (Theorem 1), whereas Collins and Roark can produce only a subset of them. This is because they rely on grammar rules and additional rules prescribing what structures are allowed in parse trees (They call them "allowable chains" and "allowable triples" ). These rules are necessary for making their search space manageable, but they make it impossible to produce some trees.

**R3: How is the attention computed in the sequence-based ablation?** No attention is computed. At each step, we simply predict the target node as an integer in $[0, 249]$. It works because the rightmost chain is shorter than (or equal to) the sentence length, and all sentences in the datasets are shorter than 250.

**R4: Why does beam search help so little?** We were also surprised to find that beam search helps only marginally. A possible reason is that our method is trained with a local loss at each step, whereas prior work has demonstrated beam search works most effectively when combined with global losses. [1]

**R1, R2: The potential impact of incremental parsing on NLP.** Besides psycholinguistic motivation, incremental parsing is also useful in NLP applications. It produces a parse tree before a complete sentence is available, which is desirable when the agent responds to streaming input in real-time. For example, [2] a human-like conversational agent needs to process input information incrementally, since humans do not wait until the end of every sentence to respond.

**R2: Evaluate the method on speeches rather than texts.** This is a good idea since speech is a domain where it is more important to parse incrementally. However, that is out of the scope of this paper. Also, most existing incremental parsers [6,7,31] were evaluated on texts.

[1] Zhang and Nivre. "Analyzing the Effect of Global Learning and Beam-search on Transition-based Dependency Parsing", COLING 2012

[2] Schlangen and Skantze, "A General, Abstract Model of Incremental Dialogue Processing", EACL 2009

[Meta-Review · NeurIPS 2020]

This is a borderline paper. The technical contribution is interesting and appreciated by the reviewers. The results match the state of the art on PTB and are better on CTB. There are, however, some concerns with the paper. One of the reviewers summarized it very well: "In its present form, the scope of the paper seems too narrow. It is also somewhat unclear whom the intended audience ought to be. If the work aims to say something about psycholinguistics, the experiment should reflect that. If the work's goal is to support NLP applications, further justifications and motivations should be provided as to how a strongly incremental constituency parser might be useful in a current NLP pipeline. If the work aims to shed lights on our understanding of GNN, the paper would need to be refocused accordingly." It is indeed unclear what the impact of the paper in its current form is. The authors claim that the main novelty with respect to other incremental parses is the action space, but the ablation experiments show that the attach-juxtapose transitions have a much smaller impact on performance compare do the GNN, at least on PTB (maybe the impact on CTB is larger?). The authors acknowledge that incremental parsing would indeed be more important in speech than in text, but then consider evaluation on speech to be future work. The authors claim that incremental parsing would be useful in NLP applications, but for the example application, human-like conversational agent, speech would again be more appropriate than text.